# Beyond Lipid-Lowering: Effects of Statins on Cardiovascular and Cerebrovascular Diseases and Cancer

**DOI:** 10.3390/ph15020151

**Published:** 2022-01-26

**Authors:** Yoichi Morofuji, Shinsuke Nakagawa, Kenta Ujifuku, Takashi Fujimoto, Kaishi Otsuka, Masami Niwa, Keisuke Tsutsumi

**Affiliations:** 1Department of Neurosurgery, National Hospital Organization Nagasaki Medical Center, 2-1001-1 Kubara, Omura 856-8562, Japan; tsutsumi.keisuke.pn@mail.hosp.go.jp; 2Department of Pharmaceutical Care and Health Sciences, Faculty of Pharmaceutical Sciences, Fukuoka University, 8-19-1 Nanakuma, Jonan-ku, Fukuoka 814-0180, Japan; shin3@fukuoka-u.ac.jp; 3Department of Neurosurgery, Graduate School of Biomedical Sciences, Nagasaki University, 1-7-1 Sakamoto, Nagasaki 852-8501, Japan; kentaujifuku@hotmail.com; 4Division of Gerontology and Geriatric Medicine, Department of Medicine, University of Washington School of Medicine, Seattle, WA 98108, USA; t.fujimototakashi@gmail.com; 5Department of Cardiology, National Hospital Organization Nagasaki Medical Center, 2-1001-1 Kubara, Omura 856-8562, Japan; kai51005100@gmail.com; 6BBB Laboratory, PharmaCo-Cell Company Ltd., Dai-ichi-senshu bldg. 2nd Floor, 6-19 Chitose-machi, Nagasaki 852-8135, Japan; niwa@pharmacocell.co.jp

**Keywords:** cancer, cerebrovascular diseases, cardiovascular diseases, pleiotropic effect, statin

## Abstract

The 3-hydroxy-3-methylglutaryl-coenzyme A (HMG-CoA) reductase inhibitors, also known as statins, are administered as first-line therapy for hypercholesterolemia, both as primary and secondary prevention. Besides the lipid-lowering effect, statins have been suggested to inhibit the development of cardiovascular disease through anti-inflammatory, antioxidant, vascular endothelial function-improving, plaque-stabilizing, and platelet aggregation-inhibiting effects. The preventive effect of statins on atherothrombotic stroke has been well established, but statins can influence other cerebrovascular diseases. This suggests that statins have many neuroprotective effects in addition to lowering cholesterol. Furthermore, research suggests that statins cause pro-apoptotic, growth-inhibitory, and pro-differentiation effects in various malignancies. Preclinical and clinical evidence suggests that statins inhibit tumor growth and induce apoptosis in specific cancer cell types. The pleiotropic effects of statins on cardiovascular and cerebrovascular diseases have been well established; however, the effects of statins on cancer patients have not been fully elucidated and are still controversial. This review discusses the recent evidence on the effects of statins on cardiovascular and cerebrovascular diseases and cancer. Additionally, this study describes the pharmacological action of statins, focusing on the aspect of ‘beyond lipid-lowering’.

## 1. Introduction

Statins potently inhibit 3-hydroxy-3-methylglutaryl-coenzyme A (HMG-CoA) reductase by competitively blocking the active site of the enzyme. Statins decrease cholesterol biosynthesis and thereby reduce plasma cholesterol levels. The development of statins as cholesterol-lowering agents began in the mid-1970s when they were discovered as a fungal metabolite, with the first of these being a natural product called mevastatin [1]. Since this discovery, vigorous efforts have been made to develop novel statins, leading to the introduction of a total of eight varieties to date. Over four decades of use have led statins to become one of the most widely prescribed drugs globally, especially for cardiovascular diseases [2,3,4,5,6,7,8]. The association between dyslipidemia and cardiovascular disease has been comprehensively established. On the other hand, statins exhibit pleiotropic properties that are independent of their lipid-lowering effects [9]. Independent of lipid-lowering effect, statins have been suggested to inhibit the development of cardiovascular disease through anti-inflammatory, antioxidant, vascular endothelial function-improving, plaque-stabilizing, and platelet aggregation-inhibiting effects [10,11,12,13,14,15]. The preventive effect of statins on atherothrombotic stroke is well established, but statins can influence other cerebrovascular diseases. Thus, statins have many neuroprotective effects in addition to lowering cholesterol. Furthermore, research suggests that statins cause pro-apoptotic, growth-inhibitory, and pro-differentiation effects in various malignancies [16]. Preclinical and clinical evidence suggests that statins inhibit tumor growth and induce apoptosis in specific cancer cell types. The pleiotropic effects of statins on cardiovascular and cerebrovascular diseases have been well established; however, the effects of statins on cancer patients have not yet been fully elucidated and are still controversial. This review discusses the recent evidence on the effects of statins on cardiovascular and cerebrovascular diseases and cancer, in addition to the pharmacological action of statins, focusing on the aspect of ‘beyond lipid-lowering’.

## 2. Pharmacological Action of Statins

Cholesterol is an essential component to maintain life, serving as a basic ingredient for cell membranes, steroid hormones, and bile acids. It is categorized into exogenous cholesterol, which is ingested via food, and endogenous cholesterol, which is primarily synthesized in the liver, with the latter being predominant in the human body. The cholesterol biosynthesis pathway involves multistage reactions, where the rate-determining step is the reaction to synthesize mevalonate from HMG-CoA. This rate-determining step is catalyzed by HMG-CoA reductase, which expression is subject to feedback control by intracellular cholesterol content. The homeostasis of intracellular cholesterol is maintained through this feedback control [17]. Statins are indispensable components for the treatment of dyslipidemia and the prevention of cardiovascular diseases due to their ability to suppress cholesterol biosynthesis by inhibiting HMG-CoA reductase.

The first statin, compactin (mevastatin), was discovered by Endo et al. in 1973 within the culture medium of *Penicillium citrinum* [18]. Since this discovery, vigorous efforts have been made to develop novel statins, leading to the introduction of eight statin varieties to date. Although these statins commonly possess a structure similar to HMG-CoA, their cholesterol-lowering effect and pharmacokinetics differ [1]. Clinically employed statins include lipophilic simvastatin, atorvastatin, pitavastatin, and fluvastatin, and hydrophilic pravastatin and rosuvastatin. While lipophilic statins are absorbed into cells via passive diffusion, hydrophilic statins require organic anion transport proteins (OATPs) to become absorbed into the liver. Except for fluvastatin, statins serve as substrates for OATP1B1, which gene polymorphism is reportedly responsible for the varying pharmacokinetics of statins [19]. Cytochrome P450 (CYP) is involved in the metabolism of some statins, with simvastatin and atorvastatin primarily being metabolized by CYP3A4 and fluvastatin being metabolized by CYP2C9. However, other statins are scarcely metabolized by CYP [20]. Generally, statins are well-tolerated and seldomly cause serious adverse events. However, one noteworthy adverse reaction is myopathy, which may progress to fatal rhabdomyolysis [21]. Therefore, when high statin doses are administered for treatment or when used concomitantly with drugs that inhibit CYPs or OATPs, caution should be exercised to avoid increased risks for adverse reactions.

The cholesterol-lowering effect of statins is primarily attributed to a compensatory increase in the expression of low-density lipoprotein (LDL) receptor due to the suppressed endogenous cholesterol synthesis rather than a decrease in intracellular cholesterol biosynthesis. The increased LDL receptor level, in turn, promotes hepatic uptake of blood LDL-cholesterol (LDL-C), thereby reducing its blood concentrations. This regulation of the LDL receptor expression by intracellular cholesterol content is mediated by a transcriptional regulatory mechanism comprising sterol regulatory element (SRE) and one of its binding proteins, SRE-binding protein 2 (SREBP-2) [22,23]. Sterol regulatory element-binding protein (SREBP)-2 forms a dimer with SREBP cleavage-activating protein (SCAP), which is a cholesterol sensor, on the endoplasmic reticulum membrane. Under cholesterol-rich conditions, the SREBP-SCAP complex forms a trimer with an insulin-inducing gene (*INSIG*) and remains on the endoplasmic reticulum membrane. Conversely, when intracellular cholesterol content decreases, the SREBP-SCAP complex detaches from *INSIG* and migrates to the Golgi apparatus. Subsequently, the N-terminus of SREBP-2 is cleaved by proteases (S1P and S2P) in the vicinity of the membrane-binding site [24]. The cleaved SREBP migrates into the nucleus and binds to SRE as a transcriptional factor, thereby promoting the transcription of various genes, including those of the LDL receptor and HMG-CoA reductase (Figure 1).

In the presence of cholesterol, which regulates the expression of the SREBP activation pathway and the LDL receptor gene, the SREBP-SCAP complex binds to INSIG on the endoplasmic reticulum membrane. When intracellular cholesterol decreases, the SREBP-SCAP complex detaches from INSIG and migrates into the Golgi apparatus. Subsequently, SREBP is cleaved by Site-1 protease (S1P) and Site-2 protease (S2P). The SREBP that migrated into the nucleus binds to sterol regulatory element (SRE) and promotes the transcription of the target gene.

The suppressive effects of statins on cardiovascular diseases have been demonstrated in numerous studies. Some of these studies have reported that the suppressive effects of statins on cardiovascular diseases are not necessarily correlated with the cholesterol-lowering effect; hence, it is also necessary to pay attention to the effects of statins other than the cholesterol-lowering effect (pleiotropic effects) [25,26]. Such pleiotropic effects include cardiovascular effects, such as antioxidative, antithrombotic, and functional improvement effects on endothelial cells, and non-cardiovascular effects, such as anti-inflammatory and anticarcinogenic effects. Multiple studies have indicated that pleiotropic effects of statins are partially attributed to the statin-mediated suppression of isoprenoid synthesis [27,28]. Isoprenoids are intermediates of the cholesterol biosynthesis pathway. Thus, statins suppress not only the synthesis of cholesterol but also that of isoprenoids.

Isoprenoids are critical molecules in maintaining cell functions, such as electron transport and protein functional regulation. The activity of Rho and Ras families of low molecular weight G-proteins (small G-proteins) is modulated through post-translational modification with isoprenoid biosynthesis intermediate metabolites, such as farnesyl pyrophosphate (FPP) and geranylgeranyl pyrophosphate (GGPP) [29]. The geranylgeranylation of Rho family proteins by GGPP modulates their activity by determining their intracellular localization. Statins suppress Rho activation by inhibiting GGPP production, thereby regulating the expression of vasoactive substances, such as endothelin-1 [30], vascular endothelial growth factor [30], angiotensin II receptor type-1 [31], and endothelial nitric oxide synthase (eNOS) [32]. Additionally, the activity of transcriptional factors, such as nuclear factor-kappa B, Kruppel-like factor 2, and peroxisome proliferator-activated receptor, is regulated as part of the expression mechanism underlying the anti-inflammatory effect of statins [33,34,35]. Furthermore, Rac, which belongs to the Rho family of proteins, is involved in the activation of nicotinamide adenine dinucleotide phosphate oxidase and the production of superoxides. Therefore, the statin-mediated suppression of Rac activation may stabilize eNOS and decrease oxidative stress [36,37]. Meanwhile, the FPP-mediated farnesylation of Ras regulates Ras activation by controlling its migration from the cytoplasm to the cell membrane. Ras has been revealed to activate certain intracellular signaling pathways, such as MAP kinase and PI3K pathways, and plays a pivotal role in the regulation of cell proliferation, movement, death, and other cellular phenomena. Several previous reports have indicated that statin-mediated Ras suppression regulates cellular proliferation and cell death, consequently leading to an antitumor effect [38,39]. Additionally, statins exhibit the antitumor effect by inhibiting Hippo-Yap/TAZ pathway, which is involved in cellular proliferation [40,41,42].

As mentioned above, statins suppress the production of intermediate metabolites of isoprenoid biosynthesis, such as GGPP and FPP, by inhibiting the cholesterol biosynthesis pathway, consequently suppressing the activation of small G-proteins. Small G-proteins play important roles in many systems that regulate cellular functions, and these regulatory effects are partially attributed to the pleiotropic effects of statins (Figure 2).

Statins, which inhibit 3-hydroxy-3-methylglutaryl-coenzyme A (HMG-CoA) reductase, suppress not only the biosynthesis of cholesterol but also that of isoprenoids. Low molecular weight G-proteins (small G-proteins) of the Rho and Ras families are subject to activation modulation through post-transcriptional modification with isoprenoid biosynthesis intermediate metabolites, such as farnesyl pyrophosphate (farnesyl-PP) and geranylgeranyl pyrophosphate (geranylgeranyl-PP). Small G-proteins regulate cellular proliferation, cellular differentiation, gene expression, and cell movement, among other processes. These regulatory effects are partially attributed to the pleiotropic effects of statins.

There are several possible biological mechanisms that might explain the association between statins and cardiovascular and cerebrovascular diseases and cancer. Possible mechanisms of statins on cardiovascular and cerebrovascular diseases and cancer are listed in Table 1.

## 3. Statins and Cardiovascular Diseases

An association between dyslipidemia and cardiovascular diseases has been comprehensively established [2,3,4,5,6,7,8]. Evidence on LDL-C and cardiovascular disease is more abundant than other dyslipidemias. Many epidemiological studies in Europe and the United States, including the Framingham study, have shown that higher LDL-C levels increase the incidence and mortality of coronary artery disease [55].

Statins are HMG-CoA reductase inhibitors that reduce cholesterol synthesis in the liver. By reducing intracellular cholesterol, statins increase the expression of LDL receptor on the surface of the liver. As a result, LDL uptake from the blood to the liver is increased, and plasma levels of other ApoB-containing lipoproteins, including LDLs, chylomicrons, very low-density lipoproteins (VLDLs), lipoprotein (Lp) (a), and intermediate-density lipoproteins, are decreased. Since the 1990s, large-scale clinical trials, conducted mainly in Europe and the United States, have demonstrated that lipid-lowering therapy with statins reduces cardiovascular events. In 2005, Cholesterol Treatment Trialists’ (CTT) collaborators reported that a one mmol-reduction (38.7 mg/dL) of LDL-C reduced major cardiovascular events (that include non-fatal myocardial infarction (MI), coronary heart disease death, coronary revascularization, and stroke) by 21%, regardless of the baseline LDL-C value [56]. This trial is a meta-analysis of 14 RCTs comparing statins and placebo groups, showing the efficacy of statins for cardiovascular diseases. The drugs used in this study included simvastatin, lovastatin, pravastatin, fluvastatin, and atorvastatin. In many cases, standard statins were used. Moreover, in 2010, CTT collaborators reported that a meta-analysis of all 26 randomized trials showed similar results to those in 2005. It showed a 22% suppression and 10% reduction in total mortality. In addition, more intensive statin regimens resulted in a 15% greater reduction in major cardiovascular adverse events than those given in less intensive regimens. In other words, it was suggested that the hypothesis ‘the lower, the better’ is correct to reduce the risk of cardiovascular events for LDL-C [57]. In the 2019 ESC/EAS guidelines for the management of dyslipidemias, recommendations for patients with very high risk atherosclerotic cardiovascular disease (ASCVD), diabetes mellitus (DM) with target organ damage, severe chronic kidney disease (CKD), a calculated systematic coronary risk evaluation of >10% for 10-year risk of fatal cardiovascular disease (CVD), or familial hypercholesterolemia (FH) with ASCVD risk factor, are an LDL-C reduction of >50% from baseline and an LDL-C goal of <1.4 mmol/L (<55 mg/dL). Then, recommendations are also for an LDL-C reduction of >50% from baseline and an LDL-C goal of <1.8 mmol/L (<70 mg/dL) for patients with high risk, an LDL-C goal of <2.6 mmol/L (<100 mg/dL) for moderate risk, and an LDL-C goal of <3.0 mmol/L (<116 mg/dL) for low risk [58].

Different types of statins have different degrees of LDL-C reduction, and different statin doses have different rates of this reduction. High-intensity regimens are defined as doses of statins that reduce LDL-C by 50% on average, and medium-intensity therapy is defined as doses that reduce LDL-C by 30–50% [59]. However, the degree of statin-induced LDL-C reduction varies from person to person and is affected by genetic background and medication compliance. Some people cannot tolerate and continue taking appropriate doses, requiring a change to a non-statin agent to improve dyslipidemia. Moreover, statins improve hypertriglyceridemia and reduce triglyceride (TG) levels by 10–20% of baseline values. Particularly, strong statins (rosuvastatin, pitavastatin, and atorvastatin) have a high TG-lowering effect [60]. The mechanism of the TG-lowering effect of statins is unclear, but it seems that an increase in lipoprotein metabolism is involved. It is said that an increase in the VLDL uptake rate in hepatocytes and a decrease in VLDL production rate are involved. The rate of VLDL decrease may depend on VLDL concentration before the treatment [61]. In a meta-analysis, statin dose changed the degree of high-density lipoprotein-cholesterol (HDL-C) level elevation. Statin-induced changes in HDL-C correlated positively and significantly with those of ApoA-I. In contrast to the relationships between changes in HDL-C LDL-C, there is a clear relationship between statin-induced increases in HDL-C and reductions in plasma TG [62]. In 2016, Ford and colleagues analyzed the data of the West of Scotland Coronary Prevention Study and verified the efficacy of statins for non-DM patients with a 10-year risk of ASCVD < 7.5%. With a duration of over 20 years, an 18% risk reduction of all-cause death was shown [63]. Although statins are effective for preventing ASCVD in adults aged 75 years and older, subgroup analysis of heart failure and hemodialysis patients has failed to show their effectiveness [57].

It has also been suggested that the preventive effect of statins for ASCVD may not be limited to lowering cholesterol. In the Jupiter trial, rosuvastatin reduced cardiovascular events in patients with normal LDL-C and high C-reactive protein (CRP) [44]. Independent of LDL-C lowering, statins have been suggested to inhibit the development of cardiovascular diseases through anti-inflammatory, antioxidant, vascular endothelial function-improving, plaque-stabilizing, and platelet aggregation-inhibiting effects [10,11,12].

For patients with a high risk of cardiovascular diseases, the benefits of statin therapy outweigh the risks. The risk of statin-induced severe muscle damage, including scrotum lysis, is less than 0.1%, and the risk of severe hepatotoxicity is approximately 0.001%. The risk of newly diagnosed statin-induced DM is about 0.2% per year of treatment, depending on the underlying risk of diabetes in a study population. Statins significantly reduce the risk of atherothrombotic stroke, and thus, total stroke and other adverse cardiovascular events. To date, there is no convincing evidence on a causal link between statins and cancer, cataracts, cognitive dysfunction, peripheral neuropathy, erectile dysfunction, or tendinitis [64].

## 4. Statins and Cerebrovascular Diseases

The preventive effect of statins on atherothrombotic stroke is well established, but statins can influence other cerebrovascular diseases. Thus, statins have many neurological effects in addition to lowering cholesterol. Here, we discuss the effects of statins on cerebrovascular disease from several aspects.

### 4.1. Cerebral Infarction

Statins are strongly associated with cerebrovascular diseases, especially cerebral infarction. Many large clinical trials have been conducted, showing the positive effect of statins on stroke. Representative studies include the Cholesterol Recurrent Events (CARE) Study [65], the Long-Term Intervention with Pravastatin Ischemic Disease (LIPID) Study [66], and the Heart Protection Study (HPS) [67], all showing a reduction in the incidence of stroke or cerebral infarction. In addition, the Stroke Prevention by Aggressive Reduction of Cholesterol (SPARCL) study [68] showed the efficacy of statins in patients with stroke or transient ischemic attack. These results have been attributed to the importance of the cholesterol-lowering effect, which is the primary effect of statins [67,69]. However, recent studies have shown that cholesterol reduction in stroke is not the major factor, highlighting the importance of pleiotropic effects [70,71,72]. Statins influence intracellular signaling, improve vascular endothelial function, inhibit thrombus formation, and exert anti-inflammatory and antiangiogenic effects. Statin treatment is essential for patients with carotid artery stenosis, as the pleiotropic effect stabilizes the carotid atherosclerotic plaque [73]. This finding has been confirmed by the results of the Japan Statin Treatment Against Recurrent Stroke (J-STARS) study, which showed that low-dose statin reduces the occurrence of stroke due to larger artery atherosclerosis [74]. Although statins have been well studied for the primary and long-term secondary prevention of stroke, their use in the acute phase is controversial. Large retrospective studies have shown that early resumption of statins contributes to improved survival in patients using statins prior to stroke onset [75]. On the other hand, the benefit of statin in the acute phase of stroke was not clear in the Administration of Statin on Acute Ischemic Stroke Patient Trial (ASSORT) [76]. However, recent advances in stroke treatment have supported the usefulness of statins as an early intervention. A systematic review and meta-analysis of intravenous thrombolysis, one of the core components of early treatment, has shown that post-treatment use of statins is associated with reduced intracerebral hemorrhage and mortality. Furthermore, in mechanical thrombectomy, preoperative and long-term postoperative use of statins has been suggested to reduce the risk of arterial re-occlusion [77,78,79]. In addition, experiments on animals have shown that statins can protect arterial intimal damage after endovascular mechanical thrombectomy using stent retrievers [80]. The pleiotropic properties of statins suggest that they provide benefits in many other aspects besides post-stroke prevention. The anti-inflammatory and neuroprotective effects of statins can reduce the infarct volume, findings that are supported by a me-ta-analysis of stroke imaging analysis. Suppressing oxygen glucose deprivation-induced activated microglial cells and reticulum stress by autophagy inhibition of statins are associated with the result [81,82,83]. The inhibition of aquaporin 4 (AQP4) expression by statins has also been shown to contribute to the reduced volume of cerebral infarction by suppressing brain edema [84]. In addition, recent studies have shown that the neuroinflammatory suppressive effect of statins can inhibit early and long-term epileptic seizures after stroke [85,86,87]. Therefore, statin therapy should not be withdrawn in patients previously taking statins, and patients not previously treated with statins should start receiving statins from the early stage of stroke.

### 4.2. Intracerebral Hemorrhage

The effect of statins on intracerebral hemorrhage (ICH) has been controversial, as previous epidemiological studies have shown that hypocholesterolemia may cause increasing ICH. Also, the SPARCL study showed an increased occurrence of a cerebral hemorrhage. This is related to the facts that cholesterol is important for maintaining the structure of blood vessels and that statins suppress platelet aggregation. However, recent studies have shown that statins do not increase the risk of hemorrhage, further suggesting their beneficial effects in ICH. A large meta-analysis of 42 trials revealed no apparent association between statins and risk of ICH and showed a reduction in stroke and cerebral infarction [88]. Although several studies have been conducted on statin use and cerebral microbleeds (CMBs) formation, which are related to ICH, there is no clear association [89,90]. In addition, a large prospective cohort study suggested that statins might reduce the risk of ICH [91]. The multifaceted effects of statins may reduce brain damage in patients after ICH and improve their prognosis. Several animal studies have shown that statins have many neuroprotective effects, including protection of the blood–brain barrier (BBB), inhibition of inflammatory cytokines, anti-apoptotic effect, and reduction of brain edema after ICH [92,93,94,95,96]. Although the evidence for statin use in ICH remains unclear, the risks of their use have been low, and there is no need to avoid their use intensively.

### 4.3. Cerebral Aneurysm and Subarachnoid Hemorrhage

The use of statins in patients with acute subarachnoid hemorrhage has been studied extensively because of their potential effectiveness in treating cerebral vasospasm and delayed ischemic neurological deficit (DIND). However, there is still no consensus on statin use, and their clinical usefulness remains controversial. Several animal studies suggested that statins improve early brain damage and reduce cerebral vasospasm after subarachnoid hemorrhage through their anti-inflammatory, anti-apoptotic, and AQP4 expression-inhibitory effects [97,98,99]. Several clinical trials showed that statins reduce cerebral vasoconstriction and DIND [100,101,102,103], and some meta-analyses also showed this finding [104,105,106]. In contrast, there are some studies that have not shown the clear efficacy of statins [107,108]. The pleiotropic effect of statins may potentially prevent brain damage after subarachnoid hemorrhage; thus, future large-cohort studies are desirable. In addition, recent research has focused on the rupture-preventive statin effects on unruptured cerebral aneurysms. Although several retrospective studies showed that statins reduce aneurysm rupture through their anti-inflammation and endothelial protective effects [109,110], the first prospective randomized controlled trial did not show a clear significant difference [111], thus requiring additional studies.

## 5. Statins and Cancer

In addition to the cholesterol-lowering effect, statins are reported to have anti-inflammatory and antitumor effects (statin-associated pleiotrophy). Basic research suggests that statins cause pro-apoptotic, growth-inhibitory, and pro-differentiation effects in various malignancies [16]. The statins can be more reasonable and are better tolerated than traditional chemotherapeutic agents. Statins can then be investigated as to whether they can be used to prevent or treat cancer alone or in combination with other drugs. This chapter summarizes the pros and cons of the antitumor effect of statins (Table 2).

### 5.1. Effective Molecular Markers

Some authors discussed how to detect statin-vulnerable tumors. Statin-vulnerable molecular features include mesenchymal cell state, sensitizing molecular mechanisms, such as p53 mutation, t(4;14) translocation, and impaired SREBP-mediated feedback response [112], mevalonate pathway genes, the Yes-associated protein (YAP)/transcriptional coactivator with PDZ-binding motif (TAZ) transcriptional regulators [113], and cancer stem cell maintenance among others [114,115,116].

### 5.2. Clinical Studies

Clinical studies of statin use for patients with malignant tumors are summarized in Table 3.

### 5.3. Clinical Studies

#### 5.3.1. Breast Cancer: One of the Promising Scenarios

Metabolic syndrome, including hypercholesterolemia, can harm the prognosis of breast cancer patients [139]. Although meta-analyses did not necessarily demonstrate the antitumor effect of statins against breast cancer [140], some nationwide cohort studies supported the protective effect of statins regarding breast cancer-related incidence and mortality [141,142]. Eliminating the possibility of immortal time bias and selection bias (see Limitations below), Nowakowska et al. have reported that statins used for triple-negative breast cancer (TNBC) improve overall survival (OS) in stage I to III patients [143]. Estrogen receptor (ER)-negative breast cancer cells are sensitive to statin exposure [144]. An increase of mesenchymal cell marker, vimentin, or decrease of the epithelial marker, E-cadherin, is sensitive to statins [145,146]. It is speculated that epithelial–mesenchymal transition-inducing cells are highly sensitive to statin treatment, which may suppress the metastatic potential of breast cancer [112,147]. In a phase II study, fluvastatin reduced proliferation and increased apoptosis in women with ER-negative high-grade breast cancer [117]. High-dose atorvastatin induces anti-proliferative effects through up-regulation of tumor suppressor p27 (cyclin-dependent kinase 1B) and down-regulation of oncogene cyclin D1 in phase II study of 42 patients with breast cancer [118].

There are several phase II randomized clinical trials (RCTs) that have investigated statin use in cancer. The chemo-sensitizing effect was investigated in 82 metastatic breast cancer patients, but carboplatin and vinorelbine plus simvastatin did not improve objective response rate (ORR), toxicity, and OS compared with carboplatin and vinorelbine alone. High-sensitivity CRP (hsCRP) and lactate dehydrogenase (LDH) are described as prognostic factors in breast cancer patients [119]. A trend for better activity and tolerability is observed in 66 patients with locally advanced breast cancer; however, fluorouracil, adriamycin, and cyclophosphamide (FAC) plus simvastatin did not statistically improve ORR and OS compared with FAC alone. Human epidermal growth factor receptor-related 2 (HER2) expression is the factor related to therapeutic response in that study [120]. However, the study populations were not stratified with molecular markers. Statin safety has been partially warranted concerning skin toxicity and cardiotoxicity protection in RCTs on breast cancer patients [148,149,150]. Therefore, statin administration is a promising approach for tumors with effective molecular markers and without sufficient treatment options, such as ER-negative breast cancer and TNBC. Statins have also been speculated to be effective against ER-positive breast cancer for other reasons [113]. Several phase III prospective RCTs can be searched at clinicaltrials.gov, accessed on 24 December 2021.

#### 5.3.2. Leukemia

A meta-analysis suggested the preventive effect of statins for leukemia and non-Hodgkin lymphoma [151]. In a phase I study, idarubicin and high-dose cytarabine plus pravastatin were administered in 15 newly diagnosed and 22 salvage patients with acute myeloid leukemia (AML) harboring unfavorable or intermediate prognosis cytogenetics. Compared with an historical group, complete remission was obtained in 11 of 15 new patients and 9 of 22 salvage patients. These are encouraging response rates [121]. In a phase II trial, idarubicin, cytarabine, and pravastatin improved the response rate compared with historical control in relapsed AML [122]. However, the chemotherapy plus pravastatin regimen did not meet the prespecified efficacy criteria in 24 patients with newly diagnosed AML [123].

#### 5.3.3. Multiple Myeloma

Meta-analyses suggested the preventive effect of statins for multiple myeloma [152,153]. A pilot phase II trial revealed that six patients with refractory multiple myeloma to whom simvastatin was administered showed a reduction of chemotherapy resistance compared to 10 patients without simvastatin use [124]. Lovastatin plus thalidomide–dexamethasone prolonged OS and progression-free survival (PFS) compared to thalidomide–dexamethasone alone in a phase II study [125].

#### 5.3.4. Esophageal Cancer

Meta-analysis findings suggested the pleiotropic effect of statins in esophageal cancer [154,155]. An RCT supported the feasibility of the one-year simvastatin administration for patients with esophageal cancer who had undergone esophagectomy but did not conclude the survival outcomes [126]. Statins may also have a protective effect for acute lung injury after esophagectomy [156].

#### 5.3.5. Gastric Cancer

Meta-analysis findings presented statin-associated pleiotropy in gastric cancer [157,158,159]. The effect of statins revealed the same tendency even when the effect of *Helicobacter* eradication was considered [160]. In several phase II or III RCTs, lovastatin with ubiquinone was ineffective for patients with advanced gastric adenocarcinoma [127], epirubicin, cisplatin, and capecitabine plus pravastatin did not improve PFS at six months compared with the chemotherapy alone [128], and capecitabine and cisplatin plus simvastatin did not increase PFS compared with capecitabine and cisplatin alone in advanced gastric cancer [129]. Statins were reported to increase the eradication rate of *Helicobacter pylori* in RCTs, which is favorable for gastric cancer prevention [161,162].

#### 5.3.6. Colorectal Cancer

Epidemiological studies suggested the pleiotropic effect of statins on colorectal cancer [163,164]. A retrospective cohort study revealed that preoperative statin therapy displays a strong association with reduced postoperative mortality following surgical resection for rectal cancer [165]. On the other hand, statin use at the time of diagnosis was not associated with increased PFS and OS in KRAS-mutant patients treated with chemotherapy and bevacizumab plus cetuximab [166]. In a prospective observational study, statin use during and after adjuvant chemotherapy was not associated with improved OS in patients with stage III colon cancer after curative resection, regardless of KRAS mutation status [167]. Leucovorin, 5-fluorouracil, irinotecan (FOLFIRI)/capecitabine, and irinotecan (XELIRI) plus simvastatin did not increase OS and PFS compared with FOLFIRI/XELIRI alone in metastatic colorectal cancer in a phase III RCT [130]. Simvastatin effect on long-course fluoropyrimidine-based preventive chemoradiotherapy is being studied in a phase II RCT [168]. The preventive effect of statins on colorectal cancer has also been investigated, but no consensus has been reached yet [169].

#### 5.3.7. Hepatocellular Carcinoma

A meta-analysis described statin-associated pleiotropy in hepatocellular carcinoma [170,171,172]. There are several phase II or III RCTs that studied this effect. Sorafenib plus pravastatin did not improve time to progression (TTP), PFS, and OS compared with sorafenib alone [131,132], but improved TTP in another study [133]. In a study of transcatheter arterial embolization followed by fluorouracil, the addition of pravastatin prolonged OS compared with the standard therapy alone in advanced hepatocellular carcinoma [134]. The use of atorvastatin for preventing hepatocellular carcinoma recurrence after curative treatment is being investigated in an RCT (NCT03024684).

#### 5.3.8. Pancreatic Cancer

Epidemiological studies have suggested the pleiotropic effect of statins on pancreatic cancer [46,173]. However, gemcitabine plus simvastatin did not decrease TTP compared with gemcitabine alone in advanced pancreatic cancer [135].

#### 5.3.9. Lung Cancer

Although meta-analysis did not demonstrate statin-associated pleiotropy against lung cancer [174] among lipid-lowering medication users in a nationwide study, adherence was inversely associated with reduced cancer-specific mortality in lung cancer [175]. An exploratory analysis of the Canakinumab Anti-inflammatory Thrombosis Outcome Study, in which a monoclonal antibody targeting interleukin-1-beta was studied primarily to reduce major adverse cardiovascular events, indicated that the drug is associated with a significant reduction of lung malignancy [176]. Recent epidemiological studies have also suggested that the addition of statins to tyrosine kinase inhibitors (TKI) targeting epidermal growth factor receptor (EGFR) may be effective [177,178]. Statins were associated with better clinical outcomes in malignant pleural mesothelioma and advanced non-small-cell lung cancer patients treated with programmed cell death-1 (PD-1) inhibitors in a retrospective study [179]. Standard chemotherapy plus pravastatin did not offer additional benefit compared with chemotherapy alone in patients with small-cell lung cancer [136]. Afatinib plus simvastatin did not improve response rates compared with afatinib alone in patients with non-adenocarcinomas [137], but gefitinib plus simvastatin were reported to demonstrate higher tumor response rates and longer PFS compared to gefitinib alone in patients with EGFR wild type non-adenocarcinomas [138].

#### 5.3.10. Renal Cell Carcinoma

Although a meta-analysis did not demonstrate pleiotropy of statins in renal cell carcinoma [180], the hypothesis has not been conclusive [181,182]. Statins were reported to be favorable for patients treated with sunitinib or immune checkpoint inhibitors [183,184]. The association between statin use and a reduced risk of progression and OS has been inconsistent in RCTs with patients with localized or locally advanced renal cell carcinoma after surgery [185,186].

#### 5.3.11. Bladder Cancer

Although meta-analysis findings did not demonstrate the pleiotropic effect of statin in bladder cancer [187], the hypothesis has not been conclusive [188]. A large population-based study revealed that statin users have better OS than nonusers with non-muscle-invasive bladder cancer but did not have a chemo-preventive effect [189]. The chemo-preventive effect of statin in bladder cancer remains to be investigated [190].

#### 5.3.12. Prostate Cancer

Serum cholesterol levels and metabolic syndrome may be potential risk factors for prostate cancer, but this suggestion remains inconclusive [191,192,193]. Many studies described the possibility of statin-associated pleiotropy in prostate cancer [187,194]. However, in a retrospective analysis, statin administration improved biochemical PFS but did not prolong OS in patients with prostatic cancer who had undergone radical prostatectomy [195]. A brachytherapy investigation suggested that statins, especially atorvastatin, may improve most clinical presentations with a nonsignificant improvement in 8-year biochemical PFS [196]. Postoperative treatment with atorvastatin might have contributed to the earlier recovery of erectile function after nerve-sparing radical retropubic prostatectomy [197]. The chemo-preventive effect of statin in prostate cancer remains to be investigated [198,199]. An RCT was designed to assess the potential synergies of metformin and atorvastatin for prostate cancer but has been terminated due to the low incidence of eligible patients [200].

#### 5.3.13. Malignant Melanoma

Some observational studies and secondary analysis indicated statin advantages in melanoma [201,202,203]. However, a meta-analysis of 20 RCTs of statins and fibrates for heart disease prevention found a favorable but not statistically significant effect for malignant melanoma [204]. In a randomized phase II clinical trial, a 6-month course of lovastatin for clinically atypical nevi did not induce favorable changes compared with placebo [205].

### 5.4. Limitations

There are some limitations to this study.

#### 5.4.1. Drawbacks of Epidemiological Studies

Epidemiological studies are useful and important, but the potential for selection bias and immortal time bias must be considered [206,207]. No matter how well-designed epidemiological studies are, one is unable to explain all potential sources of confounding factors and bias, and confounding factors cannot always be removed from cohort observational studies.

#### 5.4.2. Off-Label Use

In many observational and retrospective clinical studies, at least nominally, statins were prescribed for the primary prevention for patients with hypercholesterolemia or secondary prevention of coronary artery disease, stroke, and any other cardiovascular diseases. Therefore, such statin administration was for off-label use for malignancies.

#### 5.4.3. Natural History: Is Hypocholesterolemia or Hypercholesterolemia Harmful to Malignancies?

The Framingham study reported that after the age of 50 years, there is no increased overall mortality with either high or low serum cholesterol levels, and the association of mortality with cholesterol values might be confounded by the diseases predisposing to death [208]. Both hypocholesterolemia and hypercholesterolemia are reported to be harmful to malignancies or all-cause mortality [209,210,211]. The population with mildly or moderately high blood cholesterol levels (from 211 to 251 mg/100 mL, for example) tends to have a better prognosis [211]. This tendency is also observed in a statin-administered group [212]. Patients with coronary artery disease or statin-eligible hypercholesterolemia have a high incidence of cancer [213,214]. In a meta-analysis of stroke recurrence, although metabolic syndrome was associated with all-cause mortality, the role of its components, such as hypercholesterolemia, in predicting all-cause mortality was not statistically significant [215]. Additionally, about 40% of untreated patients with FH, who were carriers of the *V408M* mutation or *Afrikaner-2* mutation in exon 9 of the LDL receptor gene, reached a normal life span. At the end of the 19th century, the standardized mortality ratio of this population was lower than that of the general population [216]. Therefore, although the results of previous studies remain inconsistent, hypocholesterolemia and hypercholesterolemia probably seem to be correlated with tumors. Some cancers are reported to induce hypocholesterolemia [217], but others are not [218]. However, the causal relationships remain unknown, and further studies are needed. It should be noted that the population with mild to moderate hypercholesterolemia tends to have a favorable prognosis for the incidence and/or mortality of cancer and/or all-cause mortality.

#### 5.4.4. Do Statins and Lipid-Lowering Drugs Have Carcinogenicity?

A possibility of the carcinogenic effect of statins has been described in some observational studies on breast cancer [219], lymphoid malignancies [220], prostate cancer [221], bladder cancer [222], any malignancies [223], cancer in elderly patients [224,225], and in a meta-analysis or commentaries [212,226,227]. However, other studies found no such effect [228,229,230]. Newman and Hulley summarized that clofibrates and statins cause cancer in rodents, in some cases at levels of animal exposure close to those prescribed to humans, compared to a few antihypertensive drugs. However, this result of animal experiments could be directly extrapolated to humans, and evidence of carcinogenicity of lipid-lowering drugs from clinical trials in humans is inconclusive because of inconsistent results and insufficient duration of follow-up [231]. Recently, the Improved Reduction of Outcomes: Vytorin Efficacy International Trial (IMPROVE-IT) has demonstrated that ezetimibe, a non-statin drug inhibiting the intestinal absorption of cholesterol by targeting Nieman–Pick C1-Like 1 transmembrane protein, added to simvastatin improved the outcome of the patients with coronary artery disease [232]. However, the Simvastatin and Ezetimibe in Aortic Stenosis (SEAS) trial in patients with aortic valve stenosis showed an unexpected increase in cancer incidence during a median follow-up of 52.2 months [233]. Post-hoc analysis of the IMPROVE-IT group and the SEAS registry-based observational study after 21 months follow-up did not show an increase in cancer incidence and mortality, respectively [234,235]. The simvastatin implication remains to be clarified. The presence or absence of a harmful effect of ezetimibe on cancer is currently controversial.

As discussed above, the administration of lipid-lowering drugs may be just a confounding factor, and long-term survivors with mild to moderate hypercholesterolemia may tend to develop tumors. In short, long-term clinical trials and careful surveillance is still needed to determine whether cholesterol-lowering drugs cause cancer in humans [231].

#### 5.4.5. Do Statins and/or Lipid-Lowering Drugs Improve the True Endpoint, All-Cause Mortality?

Historically, clofibrates reduced the risk of myocardial infarction but tended to increase cancer, although not significantly, and failed to reduce all-cause mortality [227]. Recently, in the first randomized controlled study, evolocumab, proprotein convertase subtilisin-kexin type 9 (PCSK9) inhibitor, has significantly improved the recurrence of cardiovascular disease but without significant difference in all-cause mortality [236,237]. Statins have been reported to improve all-cause mortality in a large-scale meta-analysis; however, their effect on cardiovascular disease has been the best factor, and they do not always improve cancer survival [57]. Even if cancer survival is improved, statins may not necessarily improve OS compared with medical advice alone [238].

### 5.5. Perspective

Therefore, it is indispensable to verify the antitumor effect of statins in prospective controlled RCTs to clarify their true effect, as frequently pointed out by the authors of many basic, observational, and clinical studies. Their effect appears promising to stratify with molecular markers and treat in the direction of precision medicine, especially for tumors with few treatment options [112].

## 6. Conclusions

This review described the pharmacological action of statins, focusing on the aspect of ‘beyond lipid-lowering.’ Furthermore, we discussed the recent evidence on the effects of statins on cardiovascular and cerebrovascular diseases and cancer. Statins suppress the production of intermediate metabolites of isoprenoid biosynthesis, such as GGPP and FPP, by inhibiting the cholesterol biosynthesis pathway, consequently suppressing the activation of small G-proteins. Small G-proteins play important roles in many systems that regulate cellular functions, and these regulatory effects are partially attributed to the pleiotropic effects of statins. The preventive effect of statins on cardiovascular diseases and atherothrombotic stroke is well established, and is mainly due to cholesterol lowering. However, statins may have other effects that are unrelated to cholesterol-lowering, on cerebrovascular diseases. Statins have been suggested to inhibit the development of cardiovascular diseases through anti-inflammatory, antioxidant, vascular endothelial function-improving, plaque-stabilizing, and platelet aggregation-inhibiting effects. Several studies have shown that statins have many neuroprotective effects, including protection of the BBB, inhibition of inflammatory cytokines, an anti-apoptotic effect, and reduction of brain edema. Basic research suggests that statins cause pro-apoptotic, growth-inhibitory, and pro-differentiation effects in various malignancies [16]. If they are effective against tumors, the statins can be more reasonable and are better tolerated than traditional chemotherapeutic agents. Statins can then be investigated for their use in the prevention or treatment of cancer alone or in combination with other drugs. It is indispensable to verify the antitumor effect of statins in prospective controlled RCTs to clarify their true effect, as frequently pointed out by the authors of many basic, observational, and clinical studies. Although many animal models and non-randomized data on the pleiotropic effects of statins seems promising and the therapeutic efficacy of statins on cardiovascular and cerebrovascular diseases is being established, proper long term clinical trials and results are necessary to evaluate their therapeutic efficacy on cancer. It is also crucially important from the perspective of drug repositioning [239].

## Figures and Tables

**Figure 1 pharmaceuticals-15-00151-f001:**
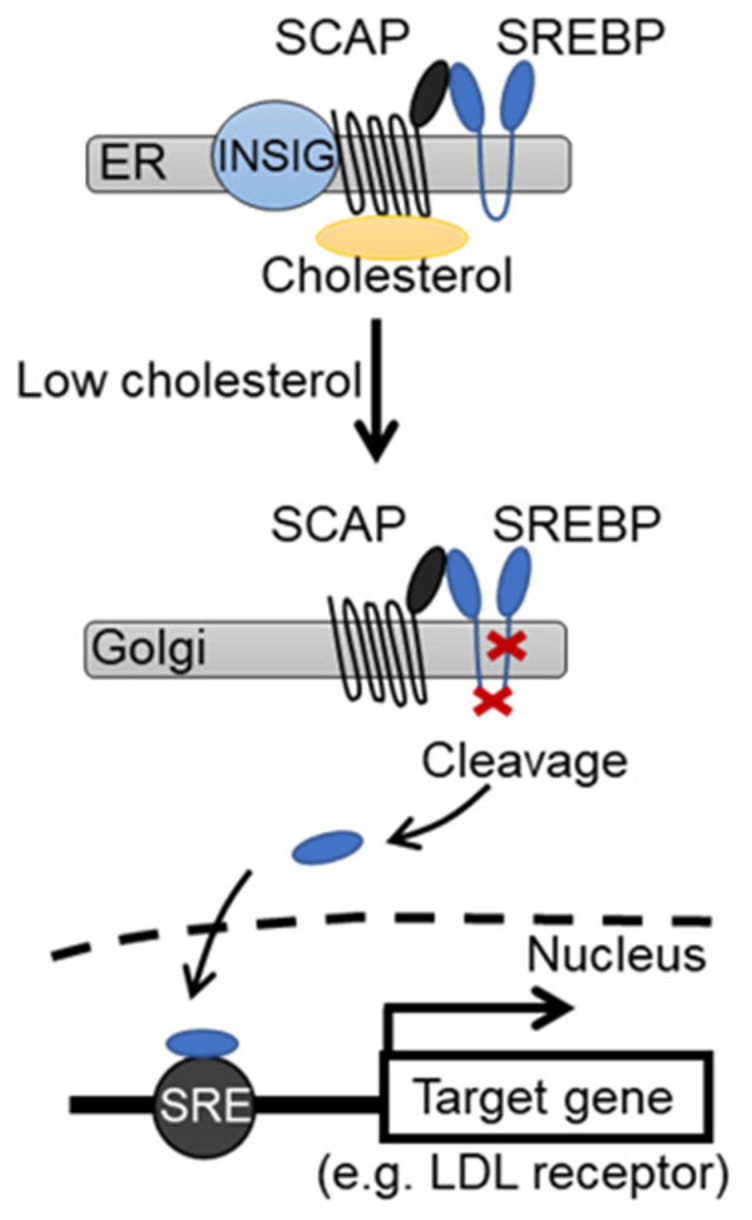
Genetic effects of cholesterol on sterol synthesis and low-density lipoprotein (LDL) receptor expression.

**Figure 2 pharmaceuticals-15-00151-f002:**
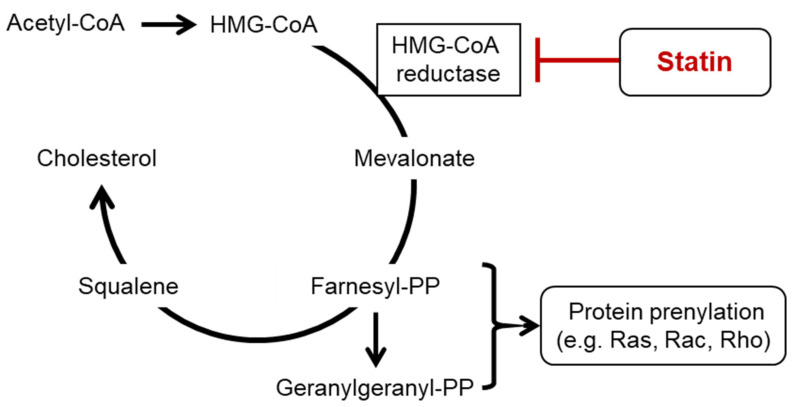
Action of cholesterol and statins on the isoprenoid biosynthesis pathway.

**Table 1 pharmaceuticals-15-00151-t001:** Possible mechanisms of statins on cardiovascular and cerebrovascular diseases and cancer.

**Lipid-lowering activities**	
Cholesterol biosynthesis ↓	[1]
LDL-receptors ↑	[22,23]
**Endothelial function**	
Expression and activity of Nitric oxide ↑	[32]
Endothelin-1 ↓	[30]
Angiotensin II receptor ↓	[31]
NF-κB activation ↓	[43]
**Anti-inflammatory effects**	
Pro-inflammatory cytokines ↓	[33,34,35]
C-reactive protein ↓	[44]
Adhesion molecules ↓	[45]
Matrix Metalloprotease ↓	[46,47]
NF-κB activation ↓	[43]
**Antioxidant activity**	
NADPH oxidase activity ↓	[37]
Reactive oxygen species production ↓	[36]
**Antithrombotic activities**	
Tissue factor expression ↓	[48]
Plasminogen activator inhibitor-1 expression ↓	[49]
Platelet activation ↓	[50]
Tissue-type plasminogen activator expression ↑	[49]
**Angiogenesis**	
Endothelial progenitor cells ↑	[51,52]
PI3 kinase activity ↑	[52]
Angiogenesis ↑	[53]
(Statins have biphasic effects on angiogenesis; high-dose statins inhibit angiogenesis)	
**Antitumor activity**	
Pro-apoptotic protein ↑	[54]
Cell proliferation ↓	[38,39]
Angiogenesis (High dose) ↓	[53]
Hippo-Yap/TAZ pathway ↓	[40,41,42]

**Table 2 pharmaceuticals-15-00151-t002:** Pros and cons of statin use for patients with malignant tumors.

Pros	Cons
Economically reasonable and well-tolerated	Off-label use
Except for hypocholesterolemia	Hypercholesterolemia
Favorable for malignancies?	Carcinogenic?
Many observational studies	Biases
Statins improve cardiovascular outcomes	Lipid-lowering drugs may not necessarily improve all causes of death

**Table 3 pharmaceuticals-15-00151-t003:** Clinical studies of statin use for patients with malignant tumors.

Authors, Year	Study Type	Patients	Evaluation	Comparison	Outcome	Results
Garwood, 2010 [117]	II	High grade ER- negative breast cancer	High dose fluvastatin	Low dose fluvastatin	Ki-67 index, caspase 3 cleavage	Fluvastatin increases apoptosis and decreases proliferation of cancer cells.
Feldt, 2015 [118]	II	Invasive breast cancer	Atorvastatin	None	p27, cyclin D1	Atorvastatin induces anti-proliferative effects through up-regulation of tumor suppressor p27 and down-regulation of cyclin D1.
Alarfi, 2020 [119]	II RCT	Metastatic breast cancer	Simvastatin, carboplatin, vinorelbine	Carboplatin, vinorelbine	ORR, OS	The chemo-sensitizing effect was investigated, but simvastatin did not improve ORR, and OS.
Yulian, 2021 [120]	II RCT	Advanced breast cancer	Simvastatin, FU, ADM, CPA	FU, ADM, CPA	ORR, OS	Simvastatin increased pathlogical ORR but did not improve OS.
Kornblau, 2007 [121]	I	New AML and recurrent AML	Pravastatin, idarubicin, cytarabine	Historical control	ORR	Pravastatin idarubicin, and high-dose cytarabine induce CR in 11 new patients and 9 salvage patients.
Advani, 2014 [122]	II	Relapsed AML	Pravastatin, idarubicin, cytarabine	Historical control	ORR	Idarubicin, cytarabine, and pravastatin improve the ORR.
Advani, 2018 [123]	II RCT	New AML	Pravastatin, idarubicin, cytarabine	Idarubicin, cytarabine	ORR	Pravastatin did not meet the prespecified efficacy criteria in newly diagnosed 24 AML patients.
Schmidmaier, 2007 [124]	II	Multiple myeloma, treated with two cycles of bortezomib or bendamustine	Simvastatin plus additional 2 cycles of bortezomib or bendamustine	Additional 2 cycles of bortezomib or bendamustine	Chemotherapy resistance	Simvastatin reduces chemotherapy resistance in 6 patients with refractory MM compared to 10 patients treated with chemotherapy alone.
Hus, 2011 [125]	II RCT	Relapsed or refractory multiple myeloma	Lovastatin, thalidomide, dexamethasone	Thalidomide, dexamethasone	OS, PFS	Lovastatin prolongs OS and PFS.
Alexandre, 2020 [126]	II RCT	Esophageal cancer	Esophagectomy with simvastatin	Esophagectomy without simvastatin	OS, PFS	The one-year simvastatin administration for patients with esophageal cancer who had undergone esophagectomy did not conclude the survival outcomes.
Kim, 2001 [127]	II	Advanced gastric cancer	Lovastatin, ubiquinone	None	ORR, toxicity	Lovastatin with ubiquinone was ineffective. NO ORR improvement was observed.
Konings, 2010 [128]	II RCT	Advanced gastric carcinoma	Pravastatin, epirubicin, cisplatin, capecitabine	Epirubicin, cisplatin, capecitabine	OS, PFS	Pravastatin did not improve OS and PFS.
Kim, 2014 [129]	III RCT	Metastatic gastric or EC junction adenocarcinoma	Simvastatin, capecitabine, cisplatin	Capecitabine, cisplatin	PFS	Simvastatin did not increase PFS compared with chemotherapy alone.
Lim, 2015 [130]	III RCT	Metastatic colorectal cancer	Simvastatin, FOLFIRI or XELIRI	FOLFIRI or XELIRI	OS, PFS	Simvastatin plus chemotherapy did not increase OS and PFS compared with chemotherapy alone.
Jouve, 2019 [131]	RCT	Advanced hepatocellular carcinoma	Pravastatin, sorafenib	Sorafenib	OS, PFS, TTP	Sorafenib plus pravastatin did not improve TTP, PFS, and OS compared with sorafenib alone.
Blanc, 2021 [132]	II RCT	Advanced hepatocellular carcinoma	Pravastatin, sorafenib	Sorafenib alone or pravastatin alone.	OS PFS	Sorafenib or pravastatin did not improve outcomes. Sorafenib is potentially effective.
Riano, 2020 [133]	II RCT	Advanced hepatocellular carcinoma	Pravastatin, sorafenib	Sorafenib	OS, TTP	Sorafenib plus pravastatin did not improve TTP compared with sorafenib alone.
Kawata, 2001 [134]	RCT	Advanced hepatocellular carcinoma	Pravastatin, embolization, FU	Embolization, FU	OS	Transcatheter arterial embolization followed by fluorouracil and pravastatin prolongs OS compared with the standard therapy alone.
Hong, 2014 [135]	II RCT	Advanced pancreatic cancer	Simvastatin, gemcitabine	Gemcitabine	TTP	Gemcitabine plus simvastatin did not decrease TTP compared with gemcitabine alone.
Seckl, 2017 [136]	III RCT	Small cell lung cancer	Pravastatin, etoposide plus cisplatin or carboplatin	Etoposide plus cisplatin or carboplatin	OS, PFS	Pravastatin did not offer additional benefits.
Lee, 2017 [137]	II RCT	Lung cancer (NSCLC, non- adenocarcinomas)	Simvastatin, afatinib	Afatinib	ORR	Simvastatin did not improve response rates. compared with afatinib alone in patients with non-adenocarcinomas
Han, 2011 [138]	II RCT	Lung cancer (NSCLC)	Simvastatin, gefitinib	Gefitinib	PFS, ORR	No outcome improvement was observed. Simvastatin increases response rates and PFS only in patients with EGFR wild type non-adenocarcinoma.

ADM, Adriamycin; AML, acute myeloid leukemia; CPA, cyclophosphamide; CR, complete remission; EC, esophageal-gastric; EGFR, epidermal growth factor receptor; ER, estrogen receptor; FOLFIRI, Leucovorin, 5-FU, and irinotecan; FU, fluorouracil; NSCLC, non-small cell lung cancer; ORR, objective response rate; OS, overall survival; PFS, progression free survival; RCT, randomized clinical trial; TTP, time to progression; XELIRI, capecitabine and irinotecan; I, phase I; II, phase II; III, phase III.

## Data Availability

Data sharing not applicable.

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
