# Peer review of "Beyond Lipid-Lowering: Effects of Statins on Cardiovascular and Cerebrovascular Diseases and Cancer"

_pharmaceuticals, 2022, doi:10.3390/ph15020151_

Round 1
Reviewer 1 Report
Overall, the review article by Morofuji et al. is well-written and of merit.
However, it would benefit from the revision of some points:
- English grammar and typos errors (e.g. “attention should be given statin effects different from the cholesterol-lowering effect”; “But statins can influence other cerebrovascular diseases probably due to independent of cholesterol lowering”; “As a result, LDL uptake from the blood is increased, and plasma levels of other ApoB-containing lipoproteins, including LDLs, chylomicrons, very low-density lipoproteins (VLDLs), lipoprotein (Lp)(a), and intermediate-density lipoproteins”; “by using statins, the rate of HDL-C increase has been shown to be less than 10%”)
- conceptual inaccuracies. The Authors state “An association between dyslipidemia and cardiovascular diseases has been established [2-8]”; this sentence is too generic since the proven association is causal. The Authors state “In 2019 ESC/EAS guide-lines for the management of dyslipidemias, as a secondary prevention, they recommended for patients with very high risk”; however, these guidelines do not refer exclusively to secondary prevention. The Authors state “Statins may increase the risk of hemorrhagic stroke in patients with cerebrovascular diseases”; this sentence is not true and should be either attenuated or avoided. The Authors state “proper long term clinical trials and results are necessary to evaluate their therapeutic efficacy on cardiovascular and cerebrovascular diseases, and cancer”; actually, I do not think it is a correct conclusion, since there is compelling evidence showing the therapeutic efficacy of statins on cardiovascular and cerebrovascular diseases.
- unclear sentences that should be reformulated (e.g. “This result is supported by a meta-analysis of stroke imaging analysis, showing that suppressing oxygen glucose deprivation-induced activated microglial cells and reticulum stress by autophagy inhibition are associated with the study”; “In October 2021, several phase III prospective RCTs can be searched at clinicaltrials.gov.”; “If they are effective against tumors, the statins can be more reasonable and are better tolerated than traditional chemotherapeutic agents.”)
- Table 1 is not well-structured. The content of the Table would be more useful to the reader if discussed in the text.
- A Table summarizing available clinical studies would be very useful.
- Some recent articles reporting additional pleiotropic effects of statins should be cited (PMID: 33568040, PMID: 32777243, PMID: 32008148)
Author Response
We wish to express our appreciation to the reviewers and editors for his or her insightful comments, which have helped us significantly improve our manuscript.
Reviewer 1
Overall, the review article by Morofuji et al. is well-written and of merit.
However, it would benefit from the revision of some points:
Response: Thank you for your comments. We have modified our manuscript according to your comments.
- English grammar and typos errors (e.g. “attention should be given statin effects different from the cholesterol-lowering effect”; “But statins can influence other cerebrovascular diseases probably due to independent of cholesterol lowering”; “As a result, LDL uptake from the blood is increased, and plasma levels of other ApoB-containing lipoproteins, including LDLs, chylomicrons, very low-density lipoproteins (VLDLs), lipoprotein (Lp)(a), and intermediate-density lipoproteins”; “by using statins, the rate of HDL-C increase has been shown to be less than 10%”)
Response: We have corrected all the sentences that you pointed out grammar mistakes and typos as below.
“attention should be given statin effects different from the cholesterol-lowering effect”
it is also necessary to pay attention to the effects of statins other than the cholesterol-lowering effect
“But statins can influence other cerebrovascular diseases probably due to independent of cholesterol lowering”
However, statins may have other effects, that are unrelated to cholesterol-lowering, on cerebrovascular diseases.
“As a result, LDL uptake from the blood is increased, and plasma levels of other ApoB-containing lipoproteins, including LDLs, chylomicrons, very low-density lipoproteins (VLDLs), lipoprotein (Lp)(a), and intermediate-density lipoproteins”
As a result, LDL uptake from the blood to the liver is increased, and plasma levels of other ApoB-containing lipoproteins, including LDLs, chylomicrons, very low-density lipoproteins (VLDLs), lipoprotein (Lp)(a), and intermediate-density lipoproteins, are decreased.
- conceptual inaccuracies. The Authors state “An association between dyslipidemia and cardiovascular diseases has been established [2-8]”; this sentence is too generic since the proven association is causal. The Authors state “In 2019 ESC/EAS guide-lines for the management of dyslipidemias, as a secondary prevention, they recommended for patients with very high risk”; however, these guidelines do not refer exclusively to secondary prevention. The Authors state “Statins may increase the risk of hemorrhagic stroke in patients with cerebrovascular diseases”; this sentence is not true and should be either attenuated or avoided. The Authors state “proper long term clinical trials and results are necessary to evaluate their therapeutic efficacy on cardiovascular and cerebrovascular diseases, and cancer”; actually, I do not think it is a correct conclusion, since there is compelling evidence showing the therapeutic efficacy of statins on cardiovascular and cerebrovascular diseases.
Response: As reviewers’ suggestions, we have modified or deleted the sentences as below.
The Authors state “An association between dyslipidemia and cardiovascular diseases has been established [2-8]”; this sentence is too generic since the proven association is causal.
An association between dyslipidemia and cardiovascular diseases has been comprehensively established [2-8].
The Authors state “In 2019 ESC/EAS guide-lines for the management of dyslipidemias, as a secondary prevention, they recommended for patients with very high risk”; however, these guidelines do not refer exclusively to secondary prevention.
In 2019 ESC/EAS guidelines for the management of dyslipidemias, they recommended for patients with very high risk.
We deleted the phrase ‘as a secondary prevention’.
The Authors state “Statins may increase the risk of hemorrhagic stroke in patients with cerebrovascular diseases”; this sentence is not true and should be either attenuated or avoided.
We completely deleted the sentence.
The Authors state “proper long term clinical trials and results are necessary to evaluate their therapeutic efficacy on cardiovascular and cerebrovascular diseases, and cancer”; actually, I do not think it is a correct conclusion, since there is compelling evidence showing the therapeutic efficacy of statins on cardiovascular and cerebrovascular diseases.
Although many animal model and non-randomized data on the pleiotropic effects of statins seems promising and the therapeutic efficacy of statins on cardiovascular and cerebrovascular diseases is being established, proper long term clinical trials and results are necessary to evaluate their therapeutic efficacy on cancer.
We have modified the whole sentence and discussed cardiovascular/cerebrovascular diseases and cancer separately.
- unclear sentences that should be reformulated (e.g. “This result is supported by a meta-analysis of stroke imaging analysis, showing that suppressing oxygen glucose deprivation-induced activated microglial cells and reticulum stress by autophagy inhibition are associated with the study”; “In October 2021, several phase III prospective RCTs can be searched at clinicaltrials.gov.”; “If they are effective against tumors, the statins can be more reasonable and are better tolerated than traditional chemotherapeutic agents.”)
Response: As reviewers’ suggestions, we have modified the sentences as below.
“This result is supported by a meta-analysis of stroke imaging analysis, showing that suppressing oxygen glucose deprivation-induced activated microglial cells and reticulum stress by autophagy inhibition are associated with the study”
, findings that are supported by a me-ta-analysis of stroke imaging analysis. Suppressing oxygen glucose deprivation-induced activated microglial cells and reticulum stress by autophagy inhibition of statins are associated with the result
“In October 2021, several phase III prospective RCTs can be searched at clinicaltrials.gov.”
Several phase III prospective RCTs can be searched at clinicaltrials.gov.
- Table 1 is not well-structured. The content of the Table would be more useful to the reader if discussed in the text.
Response: As reviewers’ suggestions, we have modified the table 2.
- A Table summarizing available clinical studies would be very useful.
Response: As reviewers’ suggestions, we have added a table summarizing clinical studies as table 3.
- Some recent articles reporting additional pleiotropic effects of statins should be cited (PMID: 33568040, PMID: 32777243, PMID: 32008148)
Response: As your suggestions, we have cited these three recent nicely presented papers.
Reviewer 2 Report
- How do statins inhibit thrombus formation and exert anti-inflammatory and antiangiogenic effects? The authors should describe their theoretical explanation about these actions of statins.
- I suggest to authors design a table summarizing the mechanism (pleitropic or not) associated with the consumption of statins for cardiovascular and cerebrovascular diseases and cancer.
- In line 102, change the lowercase letter “s” by a capital letter.
- In line 188, change the volume unit dl by dL.
- In line 227, change …I Ford analyzed… by Ford and colleagues analyzed or Ford et al analyzed.
- What is the meaning of “DM” in line 242? diabetes mellitus?
- The spacing from 317-345 lines seems to be different from the early lines.
- Font size from 391-401 lines is bigger than early sections.
- Which es the reference for information stated in lines 446-447?
Author Response
We wish to express our appreciation to the reviewers and editors for his or her insightful comments, which have helped us significantly improve our manuscript.
Reviewer 2
- How do statins inhibit thrombus formation and exert anti-inflammatory and antiangiogenic effects? The authors should describe their theoretical explanation about these actions of statins.
Response: We have added the possible mechanisms of statin as Table 1 at the end of ‘Pharmacological action of statins’ section instead of describing them in the text.
- I suggest to authors design a table summarizing the mechanism (pleitropic or not) associated with the consumption of statins for cardiovascular and cerebrovascular diseases and cancer.
Response: As reviewers’ suggestions, we added the table of possible mechanisms of statins on cardiovascular and cerebrovascular diseases and cancer.
- In line 102, change the lowercase letter “s” by a capital letter.
Response: Thank you for pointing that out, we corrected it.
- In line 188, change the volume unit dl by dL.
Response: Thank you for pointing that out, we corrected it.
- In line 227, change …I Ford analyzed… by Ford and colleagues analyzed or Ford et al analyzed.
Response: Thank you for pointing that out, we corrected it.
6.What is the meaning of “DM” in line 242? diabetes mellitus?
Response: As you pointed out, it is diabetes mellitus. We removed the abbreviation and wrote diabetes mellitus.
7.The spacing from 317-345 lines seems to be different from the early lines.
Response: Thank you for pointing that out, we corrected them.
- Font size from 391-401 lines is bigger than early sections.
Response: Thank you for pointing that out, we corrected them.
- Which es the reference for information stated in lines 446-447?
Response: As the content of the sentence in lines 446-447 (original) is ongoing clinical trial, there is no reference now. Instead of reference, we wrote the name of database and unique ID (https://clinicaltrials.gov/ct2/show/NCT03024684). We believe the readers could reach the detail of the trial.